# Study of the Trace Element Composition of Drinking Water in Almaty City and Human Health Risk Assessment

**DOI:** 10.3390/ijerph22040560

**Published:** 2025-04-03

**Authors:** Marina Krasnopyorova, Igor Gorlachev, Pavel Kharkin, Mariya Severinenko, Dmitriy Zheltov

**Affiliations:** Institute of Nuclear Physics, Ibragimov 1, Almaty 050032, Kazakhstan

**Keywords:** drinking water, heavy metals, water contamination indices, uranium

## Abstract

This research investigates the elemental composition of 78 drinking water samples collected during the summer, autumn, and winter of 2023 in different districts of Almaty city. Seasonal average concentrations and standard deviations were calculated for a range of chemical elements, including arsenic (As), beryllium (Be), cobalt (Co), cadmium (Cd), copper (Cu), lithium (Li), molybdenum (Mo), nickel (Ni), lead (Pb), selenium (Se), uranium (U), mercury (Hg), aluminum (Al), barium (Ba), chromium (Cr), iron (Fe), manganese (Mn), strontium (Sr), vanadium (V), zinc (Zn), calcium (Ca), potassium (K), magnesium (Mg), and sodium (Na), across three distinct datasets. The sites of sampling represent various categories of drinking water sources. The quality of drinking water was assessed by comparing the obtained data with current national, international, and World Health Organization (WHO) standards. Drinking water contaminant indices for the heavy metal groups were calculated and the water quality compliance with the hygienic criteria adopted in the Republic of Kazakhstan was determined. With the exception of two sampling points, the levels of non-carcinogenic risk remained below the acceptable threshold. The predominant pathway for exposure for both adults and children was identified as the oral ingestion of hazardous elements. Carcinogenic risks linked to Ni, Pb, and Cr presence in the drinking water of Almaty were identified, with risk values at the majority of sampling sites categorically classified within the “high risk” designation. No substantial differences in carcinogenic risk levels were detected between adults and children. These results underscore the necessity for enhanced water purification methodologies and ongoing surveillance to protect public health.

## 1. Introduction

Throughout the annals of history, the quality of potable water has emerged as a pivotal determinant of human health and well-being. Nonetheless, sources of drinking water are frequently compromised by the presence of toxic chemicals and heavy metals, attributed to both natural phenomena and anthropogenic activities. Furthermore, elevated levels of heavy metals in drinking water can arise from insufficient domestic water treatment systems, the application of chemical agents during purification processes, the corrosion of pipelines, the leaching of substances from water distribution infrastructure, and inadequate storage containers [1,2]. Presently, numerous developing nations grapple with this predicament, predominantly due to their constrained economic resources to adopt sophisticated heavy metal removal technologies [3].

The deleterious effects of heavy metals have become an escalating concern, influenced by evolutionary, nutritional, and ecological factors. Metals such as iron, copper, cobalt, molybdenum, manganese, and zinc are essential to human physiology in trace amounts, functioning as catalysts for enzymatic processes. Conversely, at elevated concentrations, these metals exhibit toxic properties [4]. Unregulated concentrations of heavy metals in the environment can result in detrimental health outcomes, including stunted growth and development, carcinogenesis, and neurological impairment. Moreover, exposure to specific heavy metals, including lead and mercury, may incite autoimmune disorders, wherein the human immune system erroneously targets and damages the body’s own tissues [5].

Consequently, it is imperative to conduct a thorough examination of the concentrations of heavy metals in potable water and evaluate their corresponding health implications. Numerous investigations globally have concentrated on the assessment of heavy metals in conjunction with their possible health ramifications [6,7,8]. A significant proportion of these investigations have scrutinized both the detrimental and non-detrimental facets of heavy metals. For instance, research conducted in Kampala revealed that the presence of heavy metals in drinking water could result in unacceptably high potential health threats [9]. Another investigation documented instances of oral and dermal exposure to Pb, Cd, Ni, Cr, and Fe in groundwater among both adults and pediatric populations [10], with children exhibiting heightened susceptibility. Accordingly, the research evaluated the carcinogenic risks linked to Pb, Ni, and Cr over an extended lifespan of 70 years. A study carried out in Iraq in 2019 similarly underscored both carcinogenic and non-carcinogenic impacts of heavy metals present in drinking water [11,12].

Presently, international standards for drinking water are delineated by global bodies such as the World Health Organization (WHO) and the United States Environmental Protection Agency (USEPA) [3,13,14]. Concurrently, numerous nations formulate their own national standards pertaining to the quality of drinking water. For example, Kazakhstan has instituted its own regulations governing the acceptable levels of chemical constituents in drinking water [15,16].

Health risk assessment is predominantly predicated on the juxtaposition of quantified concentrations against established thresholds for specific elements. Notwithstanding, this methodology frequently proves inadequate, as it fails to furnish comprehensive insights regarding the degree of hazard or to pinpoint the most salient pollutants arising from chronic exposure [13,14,17]. The health risk attributed to a particular heavy metal, irrespective of its concentration falling beneath the prescribed limit, is contingent upon a multitude of factors, encompassing the nature of the element, its concentration, the magnitude of human exposure, the temporal extent of exposure, and the quantity ingested [17,18,19,20,21].

The predominant guidelines employed for health risk assessment concerning human exposure to diverse pollutants are fundamentally derived from the recommendations proffered by the USEPA and WHO [17,18,19,20,21]. Consequently, risk assessment is delineated as “the systematic process of appraising the likelihood of an event transpiring and the anticipated severity of adverse health repercussions ensuing from human exposure to environmental hazards over a determined temporal framework”. The health risk assessment relevant to each potentially hazardous metal is generally anchored in a quantitative analysis of risk levels and articulated in terms of either carcinogenic or non-carcinogenic health risks [22,23].

Risk assessments yield indispensable information for policymakers aiming to safeguard public health within their respective communities [24].

In the present investigation, the elemental composition of 78 drinking water samples, collected during the winter, summer, and autumn seasons of 2023 from various districts of Almaty, was studied. Based on the data obtained, the average seasonal content of heavy metals for each sampling point, drinking water pollution indices, as well as carcinogenic and non-carcinogenic risks were calculated.

## 2. Study Area

Almaty is located in the center of the Eurasian continent, in the southeast of the Republic of Kazakhstan (Figure 1). Its geographical coordinates are delineated as 77° E longitude and 43° N latitude. Almaty is nestled within the aesthetically pleasing foothills of the Zailiysky Alatau, which represents the northernmost segment of the Tien Shan mountain range. The city encompasses an expanse exceeding 170 square kilometers and is located within the valley formed by the Bolshaya and Malaya Almatinka rivers alongside their respective tributaries, all of which emanate from the glaciers of the Zailiysky Alatau and adjacent mountain gorges. The primary sources of potable water for Almaty are derived from mountain rivers, lakes, and groundwater.

Currently, the city’s water supply system is primarily sourced from two main types of water: approximately 40% comes from mountain river water, while 60% is derived from groundwater. The water supply infrastructure includes more than 386 wells, which are located both within the city—such as the Almaty groundwater deposit—and beyond its boundaries. The main water intake facility, the Talgar Water Intake, is situated near the village of Guldala, supplying approximately 30% of the city’s water demand. This water is used to supply all thermal power plants (CHPs) for heat and hot water production, as well as to provide drinking water to the Turksib, Zhetysu, and parts of the Alatau and Medeu districts.

Numerous wells are located within the city, with depths ranging from 150 to 500 m. Groundwater serves as the primary source of drinking water for the Alatau and Auezov districts, as well as parts of the Almaly district. A significant portion of the Medeu district, along with the entire upper zone of the city, is supplied by the Medeu Filtration Station, where mountain water from the Malaya Almatinka River and its tributaries undergoes treatment. The Headwater Treatment Facilities process water from the Bolshaya Almatinka River, supplying drinking water to the Bostandyk and parts of the Almaly district.

Additionally, two other rivers, the Kargaly and Aksay, serve as sources of drinking water for the upper part of the Nauryzbay district.

To ensure high drinking water quality, modern treatment technologies are employed at the city’s water treatment facilities. These facilities utilize a multi-stage purification process, including coagulation, sedimentation, and filtration through layers of quartz sand and crushed expanded clay, effectively removing mechanical impurities. Since 2009, sodium hypochlorite, produced by the electrolysis of table salt, has been used for water disinfection instead of liquid chlorine, enhancing both safety and environmental sustainability. This comprehensive treatment system ensures that drinking water in Almaty meets national and international quality standards.

In selecting the sampling locations, we aimed to cover all urban districts while considering our analytical capabilities. This approach ensures the most comprehensive assessment of drinking water quality across different sources within the city’s water supply network. The climatic conditions within the city exhibit a distinctly continental nature, marked by pronounced temperature variations not only across different seasons, but also within the span of a single day.

For the purpose of analyzing elemental composition, a total of 26 sampling points for drinking water were systematically selected from various locations throughout Almaty (Figure 1). The figure additionally illustrates the regions associated with both groundwater and riverine sources of drinking water.

## 3. Materials and Methods

### 3.1. Sampling

Samples at each point were collected in 1 L polyethylene bottles. Before collection, the containers were washed several times with the test water and 20% HNO_3_. The collected samples were preserved by adding 3 mL of 70% HNO_3_ per liter of water, then tightly sealed and labeled with the sample number, location, and date of collection. Prior to sampling, the tap was flushed for at least 10 min to ensure the representativeness of the sample. The samples were then delivered to the analytical laboratory, where their elemental composition was analyzed within 24 h.

During the winter, summer, and autumn seasons of the year 2023, a comprehensive total of 78 drinking water samples were systematically collected from both private residences and multi-apartment complexes situated across different districts of Almaty. The sampling procedure was meticulously executed in strict accordance with relevant regulatory frameworks [25,26,27], which govern all categories of water and delineate overarching stipulations for the sampling, transportation, and preparatory storage of water specimens designated for the assessment of their composition and inherent properties.

### 3.2. Analytical Techniques

For the purpose of elemental analysis, undiluted aqueous samples were utilized. The basic analytical techniques employed at the Institute of Nuclear Physics for the assessment of the elemental composition of liquid samples include Inductively Coupled Plasma Mass Spectrometry (ICP-MS) and Inductively Coupled Plasma Optical Emission Spectrometry (ICP-OES).

The analysis via ICP-MS was executed utilizing an ELAN-9000 quadrupole inductively coupled plasma mass spectrometer (PerkinElmer Inc., Shelton, CT, USA). The root mean square deviation of the resultant signal did not surpass 6%, exhibiting a resolution that varied between 0.6 atomic mass units (amu) and 0.8 amu at 10% peak height, and encompassing a mass range extending from 2 amu to 270 amu.

The ICP-OES analysis was carried out employing an OPTIMA-8000 dual-view inductively coupled plasma optical emission spectrometer (PerkinElmer Inc., Shelton, CT, USA), characterized by an optical range spanning from 166 nm to 900 nm and a resolution of 0.008 nm at 200 nm full width at half maximum (FWHM).

In the context of ICP-MS, the isotopes subjected to analysis, and for ICP-OES, the emission lines, were selected based on criteria that facilitated a judicious equilibrium between sufficient sensitivity for elemental quantification, minimal spectral interferences, and reduced background levels.

The calibration of the ICP spectrometers was conducted utilizing diluted standard metal solutions at concentrations of 10 µg/L and 100 µg/L, which were formulated by Perkin Elmer Pure Plus (USA).

The ICP spectrometers were calibrated every 10–15 samples to ensure measurement accuracy. To mitigate instrumental drift, internal standards were incorporated into all measured samples and calibration solutions at concentrations of 5 µg/L Rh for ICP-MS and 0.25 mg/L Sc for ICP-OES. This procedure employed single-channel dispensers and volumetric glassware in compliance with [28].

The instrumental detection limit (DL) for water samples was evaluated according to the standard procedure outlined in EPA document 40 CFR 136, App. B. The detection limits for the analyzed chemical elements were as follows (µg/L): As—0.5, Be—0.03, Co—0.07, Cd—0.05, Cu—0.5, Li—0.2, Mo—0.3, Ni—0.5, Pb—0.05, Se—3, U—0.03, Hg—0.1, Al—3, Ba—0.5, Cr—0.7, Fe—0.4, Mn—0.5, Sr—0.5, V—1, Zn—0.7, Ca—10, K—15, Mg—30, and Na—10.

### 3.3. Quality Control and Assurance

The precision of the acquired results was corroborated through the examination of duplicate samples, a blank sample, and a certified reference material (CRM) specifically designed for trace metals in potable water (CRM-TMDW-500—HPS, High Purity Standards, Charleston, SC, USA). The analytical outcomes for the CRM were found to fall within the confidence interval delineated in the CRM certification. The blank sample was synthesized utilizing deionized water (conductivity ≤ 0.2 μS).

Glass volumetric dishes and plastic containers utilized throughout the analytical procedures underwent cleansing with a diluted nitric acid solution, and were subsequently rinsed with deionized water. The glassware and containers were then subjected to air-drying at ambient temperature and preserved in a sealed condition to avert contamination.

To guarantee the reproducibility of elemental analysis, duplicate sample measurements were conducted at a ratio of one duplicate for every ten analyzed water samples.

For the purpose of accuracy control, a prepared control sample comprising a known concentration of analytes was employed. Following each batch of 15 water samples, the control sample was scrutinized to affirm analytical precision. The analyte recovery coefficients for the control sample were ascertained to be within the acceptable range of 85–115%, with the established recovery range set to 80–120%.

The statistical analysis of quality control results was executed utilizing MS Excel.

### 3.4. Heavy Metal Index

In the regulatory literature of the Republic of Kazakhstan [16], the following hygienic standards for the quality of drinking water with respect to trace elements are prescribed:

The contaminants and chemical constituents are categorized into four distinct hazard classes: class 1—extremely hazardous; class 2—highly hazardous; class 3—hazardous; and class 4—moderately hazardous.

This categorization is predicated upon indicators that elucidate the diverse levels of risk that chemical compounds and elements that contaminate water present to human health. These indicators encompass toxicity, bioaccumulation potential, the capacity to induce chronic effects, and the criterion for limiting harmfulness.

Criterion 1: In the case of heavy metals identified within hazard classes 3 and 4, their concentrations must remain below the Maximum Acceptable Concentrations (MACs) for drinking water, as delineated in the regulatory framework of Kazakhstan [15,16].

Criterion 2: Should there be the presence of multiple heavy metals classified under hazard classes 1 and 2 in drinking water, which are governed by sanitary–toxicological indicators of harmfulness, the Heavy Metal Index (HMI) must not surpass the value of 1. The HMI is computed as the sum of the ratios of the measured concentration of each metal in water to its correspondent MAC, employing the following formula [15]:(1)$$HMI=\sum_{i=1}^{n}\frac{C_{i}}{MAC_{i}}\leq 1,$$
where C_i_ is the seasonal mean concentration of the *i*-th element (µg/L), MAC_i_ is the maximum acceptable concentration of the *i*-th element (µg/L), and n is the number of analyzed elements classified as hazard classes 1 and 2 whose seasonal mean concentrations in water samples exceed the method’s detection limit (DL).

Since the regulatory literature of Kazakhstan [15,16] does not provide an MAC value for uranium, the World Health Organization (WHO) guideline value of 30 µg/L was used as the reference standard for HMI calculation [3].

### 3.5. Calculation of Carcinogenic and Non-Carcinogenic Risks

To assess water quality in terms of non-carcinogenic (HI) and carcinogenic (CR) toxic effects on the human body through both oral (ingestion) and dermal (skin contact) exposure, risk assessments are widely used in contemporary research. In general, these assessments represent the ratio of the chronic daily intake (CDI) of the *i*-th element via oral and dermal exposure to a safe dose, defined by the reference dose (RfD) or slope factor (SF).

Risk assessments were conducted for two population groups—adults and children—considering daily drinking water consumption and showering. The calculations follow the methods and interpretations proposed by the United States Environmental Protection Agency (USEPA) [18,29,30,31]:

Non-carcinogenic risk (HI):(2)$$HI=\sum {HQ}_{i}$$(3)$${HQ}_{i}=\frac{{CDI}_{idermal/oral}}{Rf{Di}_{dermal/oral}}$$

Carcinogenic risk:(4)CR=∑CRi(5)$${CR}_{i}={CDI}_{idermal/oral}\times {SF}_{idermal/oral}$$
where HI is the total non-carcinogenic hazard index, HQ_i_ is the non-carcinogenic hazard quotient for the *i*-th element, CR is total carcinogenic risk index, CR_i_ is the carcinogenic risk index for the *i*-th element, CDI_i_ (dermal/oral) is the chronic daily intake (exposure dose) of the *i*-th element (µg/(kg·day)), RfD_i_ is the reference dose of the *i*-th element (µg/(kg·day)), and SF_i_ is the slope factor of the *i*-th element ((µg/kg/day)^−1^).(6)$${CDI}_{dermal}=Ci\times \frac{SA\times Kp\times ET\times EF\times ED\times CF}{BW\times AT}$$(7)$${CDI}_{oral}=Ci\times \frac{IR\times EF\times ED}{BW\times AT}$$

C_i_—concentration of the *i*-th element in water (µg/L); SA—exposed skin surface area (cm^2^):

Adult: (18,000); child: (6600) [32,33]. Kp—skin permeability coefficient (cm/h): Co, Ni: 0.004; Cr: 0.03; Pb: 0.0001; Zn: 0.0006; Fe, Mn, Al, As, Cd, Cu, Mo, U: 0.001 [32]. ET—exposure time (h/day): 0.58 [34]. CF—unit conversion factor (L/cm^3^): 1/1000 [32,33]. BW—body weight (kg): adult: 70; child: 25 [32].

AT—average time (days): dermal exposure: adult: 10,950; child: 2190; oral exposure: adult: 2550; child: 3650.IR—intake rate (L/day or kg/day): adult: 2.2 [32,34]; child: 1 [32].EF—exposure frequency (days/year): 365.ED—exposure duration (years): adult: 70 [32,34]; child: 7 [34].

The carcinogenic risk assessment for uranium ingestion through drinking water was conducted based on its radiotoxic properties using the following formula [35,36,37,38]:(8)$${CR}_{U}=C_{U}\times 0.67\times {SF}_{U}xI$$

C_U_—concentration of uranium in water (µg/L).0.67—total activity of the most common naturally occurring uranium isotopes (U-238 + U-234) (pCi/µg) [39,40].SF_U_—uranium slope factor (Risk/pCi) (Table 1).I(L)—lifetime water consumption, calculated as IR × EF × ED, which equals 56,210 L for adults and 25,550 L for children.

Risk Classification Criteria:

USEPA’s suggested non-carcinogenic risk (HI) threshold value for HI is 1 [29,30,31,32,41]: 1 < HI ≤ 5—low risk; 5 < HI ≤ 10—moderate risk; HI > 10—high risk [32,41].

The individual hazard quotient (HQ) for each element must satisfy HQ < 1 [29,30,31,32,41]. Carcinogenic risk (CR) threshold value, according USEPA’s recommendations [29,30,31,32,42,43]: CR < 1 × 10^−6^—no risk; 1 × 10^−6^ < CR < 1 × 10^−4^—risk is tolerable; CR > 1 × 10^−4^—high risk.

Elements included in the non-carcinogenic and carcinogenic risk assessment were those for which internationally recognized guidelines or research results provide reference dose (RfD) and slope factor (SF) values (Table 1).

**Table 1 ijerph-22-00560-t001:** Dermal and oral RfDs and SFs.

Element	RfD Dermal (µg/kg × day)	RfD Oral (µg/kg × day)	SF Dermal (µg/kg/day)^−1^	SForal (µg/kg/day)^−1^	References
Fe.	300.	700.	-.	-.	[32]…
Mn.	0.8.	140.	-.	-.	[32,44]…
Co.	0.015.	0.3.	-.	35,000.	[45,46]…
Cr.	0.015.	3.	20,000.	420.	[32,44,47]…
Cu.	12.	40.	42,500.	-.	[32,48]…
Mo.	1.9.	5.	-.	-.	[32,44]…
Ni.	5.4.	20.	-.	900.	[32,44,47]…
Pb.	0.42.	3.5.	-.	8.5.	[32,44,47]…
U.	-.	3.	-.	6.4 × 10^−11^(Risk/pCi).	[44,45,49]…
Zn.	60.	300.	-.	-.	[32,44]…

### 3.6. Statistical Analysis

For the purpose of statistical examination and the revealing of correlation dependencies, we used the Statistica software package, version 12.0 (StatSoft Inc., Tulsa, OK, USA). Initially developed as an independent product in 1991, it is extensively adopted across a multitude of industries for the purposes of data processing, analysis, and visualization. The software provides users with instruments for executing intricate statistical computations and modeling, rendering it an indispensable asset for researchers, analysts, and scientists.

The cluster analysis technique employed in this investigation is a multivariate statistical methodology extensively utilized for the interpretation of complex datasets and the identification of pollution sources. Cluster analysis represents a statistical approach that categorizes heterogeneous and intricate data into several groups characterized by similar attributes, assisting in determining common sources of pollution [50].

The box plot model was used to visualize the range of risk index calculation results. A cluster analysis statistical approach was applied to classify heterogeneous and complex data into several groups with similar characteristics in an attempt to identify common sources of contamination [50]. Pearson’s correlation analysis was used to determine relationships between the seasonal mean concentrations of chemical elements in drinking water, aiming to assess potential differences in their sources and water sample characteristics.

## 4. Results and Discussion

### 4.1. Heavy Metal Concentrations

The elemental composition of seventy-eight drinking water samples, collected during the winter, summer, and autumn seasons of the year 2023 from twenty-six sampling points within private residences and multi-apartment complexes in Almaty (refer to Figure 1), was analyzed with ICP-MS and ICP-OES techniques. Consequently, the concentrations of the following chemical elements were determined: As, Be, Co, Cd, Cu, Li, Mo, Ni, Pb, Se, U, Hg, Al, Ba, Cr, Fe, Mn, Sr, V, Zn, Ca, K, Mg, and Na.

Applying the obtained data, the average concentrations (mean) and corresponding standard deviations (SD) for each sampling point were computed across the three different sampling seasons. For all analyzed samples, the concentrations of As, Be, Cd, Se, and Hg did not exceed the method detection limit (DL).

Table 2 presents mean and SD for those elements whose mean concentrations exceeded the analytical method detection limit (DL). Furthermore, the table contains the Maximum Acceptable Concentrations (MACs) and hazard classifications for each chemical element, as established by the regulatory documents of the Republic of Kazakhstan [15,16], the United States of America [21], and the World Health Organization [3].

Using Equation (1), HMI was computed for each sampling point. The obtained HMI values are presented in the last column of Table 2. The spatial distribution of HMI throughout Almaty is visually illustrated in Figure 2.

As presented in Table 2, the average concentrations of all evaluated chemical elements categorized under hazard classes 3 and 4 do not exceed their MAC. Consequently, the first hygienic criterion for the quality of drinking water is satisfied across all sampling points. Nonetheless, the behavior of nickel and cobalt is alarming. The high SD values for these elements at sampling points 6 and 11 can be explained by the high concentrations of Ni and Co in drinking water samples collected during the summer season. Particularly, nickel concentrations were attained at levels of 361 µg/L at point 6 (Figure 2) and 366 µg/L at point 11, thereby exceeding the MAC for Ni by a factor of more than three. Simultaneously, such heightened concentrations of nickel and cobalt were not revealed in samples collected during winter and autumn at these points. Furthermore, adjacent summer sampling locations (points 1 and 13) did not display elevated concentrations of these elements. This observation likely indicates the presence of localized seasonal contamination impacting specific areas of the drinking water supply.

For all sampling points, the average concentrations of elements classified under hazard classes 1 and 2 do not exceed their respective MAC values, thus fulfilling the secondary hygienic criterion for drinking water quality (HMI < 1), as indicated in Table 2. However, at points 6, 11, and 17, the HMI values approach 1, signifying relatively suboptimal water quality in relation to heavy metal contamination indices. As illustrated in Figure 2, the most pristine drinking water, as assessed by the HMI indicator, is located in the upper region of the city, near the mountainous terrain, which correlates with river-based water sources.

The primary contributors to the elevated HMI values include the following: Uranium (U): Up to 62% at point 17; sodium (Na): Up to 14% at point 25; and lithium (Li): Up to 13% at point 11. Particularly alarming is the increased concentration of uranium, as this element is classified under hazard class 1. The average uranium concentrations remain consistently elevated across all sampling points, ranging from 8.0 µg/L at point 21 to 18.6 µg/L at point 17.

### 4.2. Assessment of Carcinogenic and Non-Carcinogenic Risks

The results of the non-carcinogenic hazard quotient (HQ_i_) calculations for each element are presented as follows: For adults—Figure 3a,b; for children—Figure 4a,b.

As seen in Figure 3 and Figure 4, the HQ values for each chemical element do not exceed 1 for both adults and children, regardless of the exposure route. The highest HQ values were observed as follows: Adults: Cr: HQ = 0.42 (dermal exposure), Co: HQ = 0.42 (oral exposure); children: Cr: HQ = 0.43 (dermal exposure), Co: HQ = 0.53 (oral exposure).

The total non-carcinogenic hazard indexes (HIs) for adults and children are presented in Figure 5 and Figure 6.

As seen in Figure 5 and Figure 6, the HI values remain below the acceptable threshold (HI < 1) at most sampling points, except for points 6 and 11. The primary exposure route for both adults and children is the oral ingestion of toxic elements. Only at two sampling points do the HI values fall within the low-risk category (1 < HI ≤ 5): Point 6: Adults: HI = 1.06, children: HI = 1.29; point 11: Adults: HI = 1.19, children: HI = 1.45. This is most likely due to elevated cobalt concentrations (Table 2). The main non-carcinogenic exposure route for both adults and children remains the oral ingestion of toxic elements. However, at point 4, dermal exposure exceeds oral exposure, likely due to elevated chromium concentrations (Table 2, Figure 1 and Figure 2).

The carcinogenic risk index (CR) results for adults and children are presented in Table 3.

As shown in Table 3, the CR values for all sampling points fall within the range of 10^−6^–10^−8^ for dermal exposure, indicating no risk (no risk), and between 10^−4^ (tolerable) and 10^−2^ (unacceptable) for oral exposure. The lowest CR value was observed at sampling point 21, while the highest CR value was recorded at point 6. The main contributors to the total carcinogenic risk index are the concentrations of Ni, Pb, and Cr (Figure 7 and Figure 8). In contrast, the impact of Co and U is significantly lower.

The carcinogenic risk levels for Ni, Cr, and Pb are CR_i_ > 10^−4^, while for U they fall within the range of 10^−6^ > CR_i_ > 10^−4^, and for Co they remain below CR_i_ < 10^−6^. The risk levels are ranked as follows: Pb > Cr > Ni > U > Co.

The carcinogenic risk values for Pb (CR_p_b) across different sampling points are mostly in the “high-risk” zone. In certain “extreme” locations (points 6 and 11, where the annual average concentrations of Ni exceed MAC), the risk level for Ni exceeds CR_ni_ > 10^−3^. These points also correspond to “extreme” values for Co (CR_o_ > 10^−4^). In all other samples, Co concentrations do not pose a significant risk. The “extreme” Cr values are associated with points 25 and 26, where annual average Cr concentrations are at their highest (Table 2). No significant differences in CR levels between adults and children were observed.

The detected carcinogenic risks associated with the presence of Ni, Pb, and Cr in Almaty’s drinking water are higher than in some comparable studies [51], but are not an exception [52,53]. For example, in urban areas of Wuhan, China, where the average concentrations of Ni, Pb, and Cr were 1.36 µg/L, 1.66 µg/L, and 0.68 µg/L, respectively, the carcinogenic risk level was within the range of 1 × 10^−5^ < CR < 1 × 10^−4^ [51]. In Albania [53], due to the presence of Cr, Cd, Pb, and Ni in drinking and bottled water (1.8 µg/L; 0.076 µg/L, 2.1 µg/L; 18.1 µg/L, respectively), CR values varied within the range of 10^−3^–10^−2^. In Kenya, 99% of the investigated cases [53] exceeded the acceptable CR threshold of 10^−4^ due to high Cr concentrations (0.569 mg/L).

The authors emphasize that the obtained results are preliminary and contain several uncertainties. Firstly, uncertainties arise from the applied methodology, including the generalized calculation of chronic daily intake (CDI) and the lack of precise values for RfD and SF_u_. Nevertheless, the conducted risk assessment warrants attention from relevant governmental authorities, particularly in terms of minimizing the potential health impacts of Ni, Pb, and Cr in drinking water. Although these elements play a crucial role in metabolism, their excessive concentrations can lead to adverse health effects. Long-term exposure to elevated Ni concentrations may cause immunological effects, eczema, allergic dermatitis, and other health issues [54]. Health effects associated with Pb and its compounds include neurotoxicity, developmental delays, hypertension, and impaired hemoglobin synthesis [44]. The oral ingestion of Cr (VI) primarily affects the gastrointestinal tract, in addition to immunological and hematological effects [44].

The most likely source of the described toxic elements, such as Ni and Cr [51], in drinking water is the corrosion of water supply pipes within the centralized water distribution system of the city. This issue is a common problem in many cities worldwide [51,52,55].

### 4.3. Correlation and Cluster Analysis

Pearson’s correlation analysis was used to identify correlations between the seasonal mean concentrations of chemical elements in drinking water, aiming to assess potential differences in their sources and water sample characteristics. All statistical analyses were performed using Statistica software.

The Pearson correlation coefficients (r) for chemical elements with seasonal mean concentrations exceeding the detection limit are presented in Table 4.

A value between 0 and 1 indicates a positive correlation, a value between 0 and −1 indicates a negative correlation at a significance level of *p* < 0.05 [56], and a value of zero indicates no correlation between two variables. When r > 0.7, there is a strong correlation; when r is between 0.5 and 0.7, there is a moderate positive correlation.

Based on the data from Table 4, a cluster analysis of chemical elements was conducted, and the results are shown in Figure 9. The vertical scale in Figure 9 represents the distance, calculated as 1 − r, meaning that the smaller the distance between the analyzed elements, the stronger their correlation.

As shown in Figure 9 and Table 4, strong correlations [57] were observed between the following chemical elements: Ni–Co (0.99), Mg–Sr (0.98), Sr–Li (0.86), V–Cr (0.83), Na–Li (0.84), and K–Ca (0.82). These strong correlations between certain heavy metals suggest similar sources of origin and geochemical behavior [58,59]. Accordingly, the chemical elements can be classified into three main groups: Group 1—Mn, Al, Zn, Fe, Cu, and Pb; Group 2—K, Ca, Ba, and U; Group 3—V, Cr, Na, Mg, Sr, Li, Mo, Ni, and Co. Group 1 consists of elements primarily of anthropogenic origin. Group 2 consists of elements primarily of natural origin. Group 3 appears to have a mixed origin. The presence of uranium in the natural origin group can be explained by its relatively high concentrations in surface waters of Almaty, which result from the element’s natural dispersion in surrounding mountainous rock formations [60].

Figure 10 presents the results of the cluster analysis of the studied samples, while Figure 11 shows the spatial distribution of sampling points across clusters on the city map, corresponding to Figure 10.

Based on Figure 10 and Figure 11, all sampling points can be classified into three main groups: Group 1—Sampling points 3, 21, 14, 12, 24, 20. Almost all of these points correspond to areas with surface water sources. Point 12 is located near this region. Group 2—All remaining points except for point 11. Most points in this group are located in areas with groundwater sources. Group 3—Point 11. This point forms a separate cluster due to anomalously high concentrations of Mo, Ni, and Co in the drinking water sample collected during the summer period. Thus, the cluster analysis clearly demonstrates a strong correlation between the sampling points and the type of water source.

## 5. Conclusions

The elemental composition of drinking water samples collected from 26 selected locations in Almaty during the winter, summer, and autumn of 2023 was analyzed using ICP-MS and ICP-OES methods. Based on the obtained data, mean concentrations and standard deviations were calculated for each sampling point across the three sampling periods.

Water quality and health risk assessments were conducted by comparing the obtained data with the national, international, and WHO standards. For all sampling points, the seasonal mean concentrations of hazard class 1 and 2 elements did not exceed MAC values, and the second hygienic criterion for drinking water quality was met (HMI < 1). However, at three sampling points, the HMI was close to 1, indicating low water quality concerning heavy metal contamination indices. Moreover, in water samples taken at points 6 and 11 in the summer, abnormally high contents of nickel and cobalt were observed. There are several possible explanations for this behavior of the elements: 1. The presence of localized seasonal contamination affects specific areas of the drinking water supply. 2. Contamination of water samples at the sampling stage. 3. Contamination of water samples in the analytical laboratory. 4. Measurement errors. In any case, additional studies are required to identify the causes of the observed seasonal variations.

The highest contributors to elevated HMI values were the chemical elements U, Na, and Li. In accordance with the results of the Pearson’s correlation analysis, the uranium was of natural origin, while the elements Na and Li were probably mixed. Therefore, it can be assumed that the main loading on the population through drinking water is due to the ingress of chemical elements from the environment, including as a result of the leaching of rocks in the spring and autumn. At the same time, long-term monitoring of drinking water is necessary to obtain more objective and reliable information.

When assessing carcinogenic risks, it was found that the dominant exposure pathway for both adults and children remains oral ingestion. However, at point 4, dermal exposure exceeded oral exposure due to the high Cr concentration in drinking water samples.

The CR values across all sampling points range between 10^−6^ and 10^−8^ for dermal exposure (no risk) and 10^−4^ (tolerable) and 10^−2^ (unacceptable) for oral exposure. The key contributors to the high total risk index are the concentrations of Ni, Pb, and Cr, while the influence of Co and U is significantly lower. Carcinogenic risk levels are as follows: Ni, Cr, Pb: CR_i_ > 10^−4^; uranium (U): 10^−6^ > CR_i_ > 10^−4^; cobalt (Co): CR_i_ < 10^−6^. Overall ranking of carcinogenic risk is as follows: Pb > Cr > Ni > U > Co. CR values for Pb (CR_p_b) are mostly in the “high-risk” zone. At “extreme” locations (points 6 and 11), where annual Ni concentrations exceed MAC, the risk from Ni (CR_ni_) is above 10^−3^. These same points also show “extreme” Co values (CR_o_ > 10^−4^). In all other samples, Co concentrations do not pose a significant risk. “Extreme” Cr values were found at points 25 and 26, where annual average Cr concentrations are highest. No significant differences in CR levels were observed between adults and children.

## Figures and Tables

**Figure 1 ijerph-22-00560-f001:**
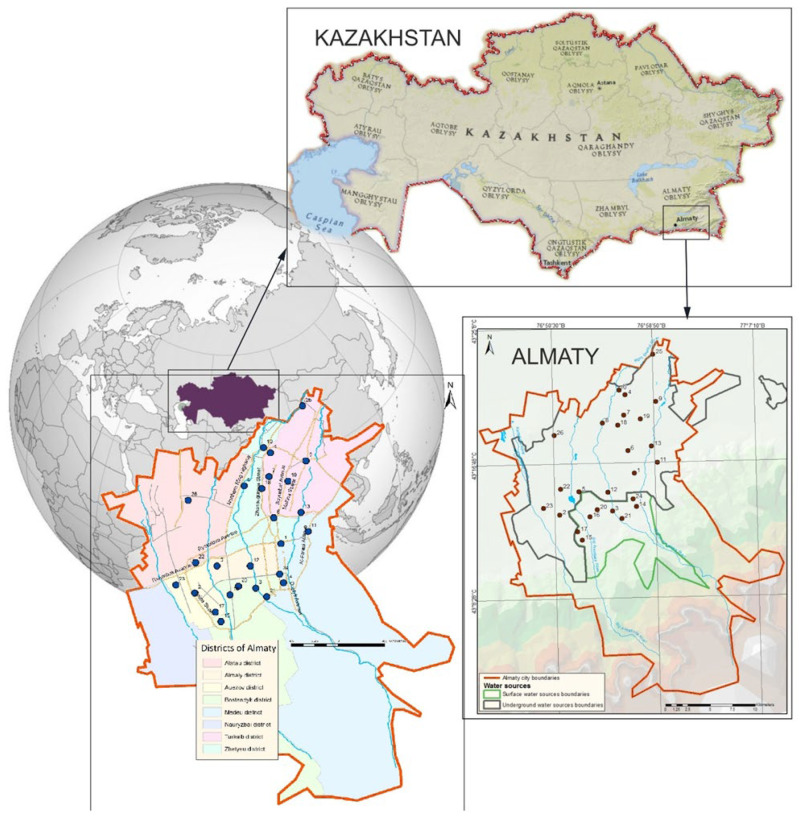
Map of the study region with designated sampling locations.

**Figure 2 ijerph-22-00560-f002:**
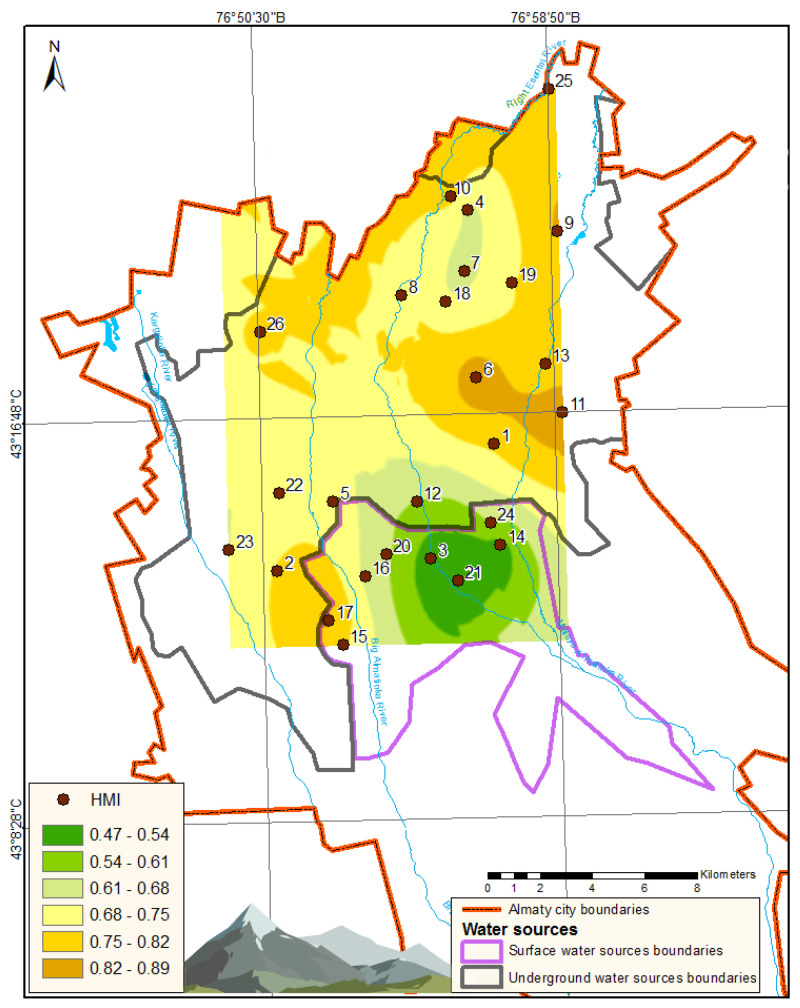
Graphical representation of HMI distribution in Almaty.

**Figure 3 ijerph-22-00560-f003:**
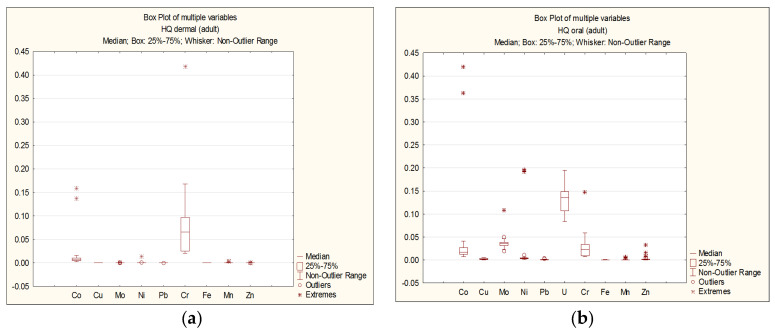
Dermal (**a**) and oral (**b**) HQi for adults.

**Figure 4 ijerph-22-00560-f004:**
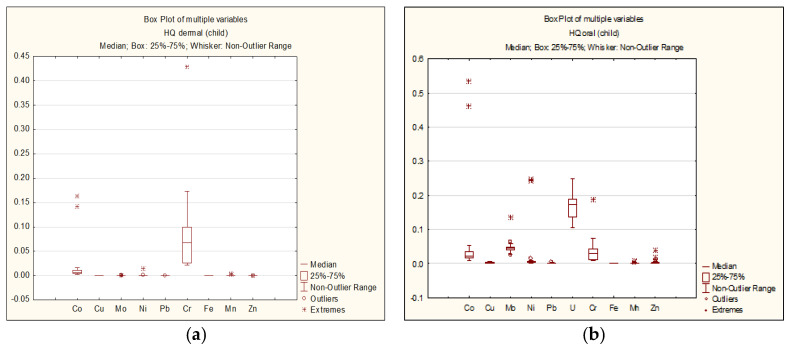
Dermal (**a**) and oral (**b**) HQi for children.

**Figure 5 ijerph-22-00560-f005:**
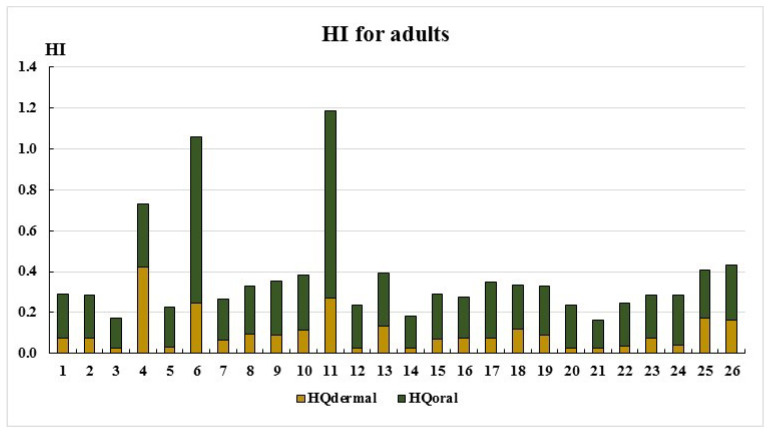
HI for adults.

**Figure 6 ijerph-22-00560-f006:**
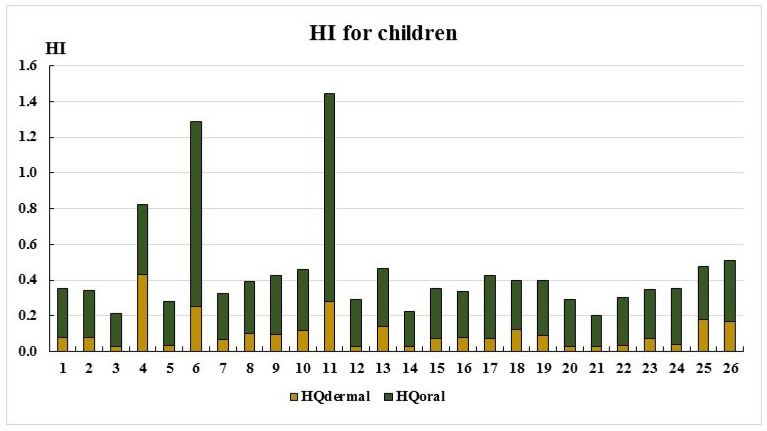
HI for child.

**Figure 7 ijerph-22-00560-f007:**
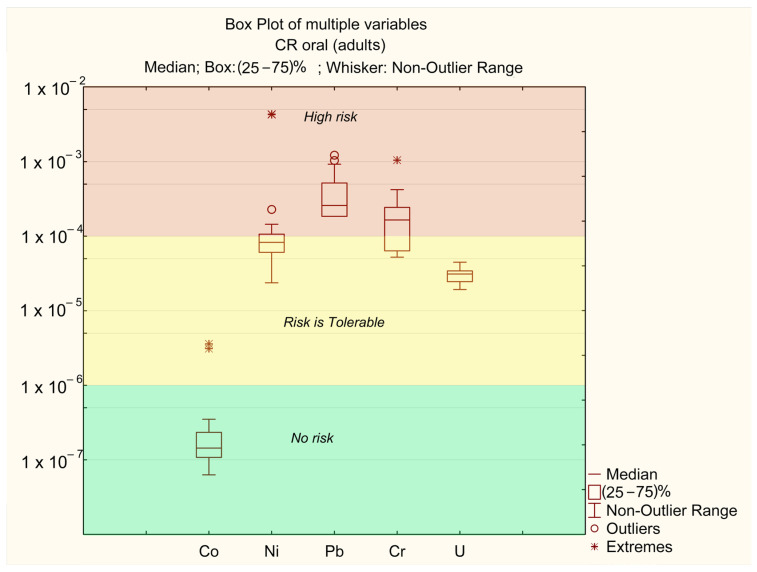
Oral CFi for adults.

**Figure 8 ijerph-22-00560-f008:**
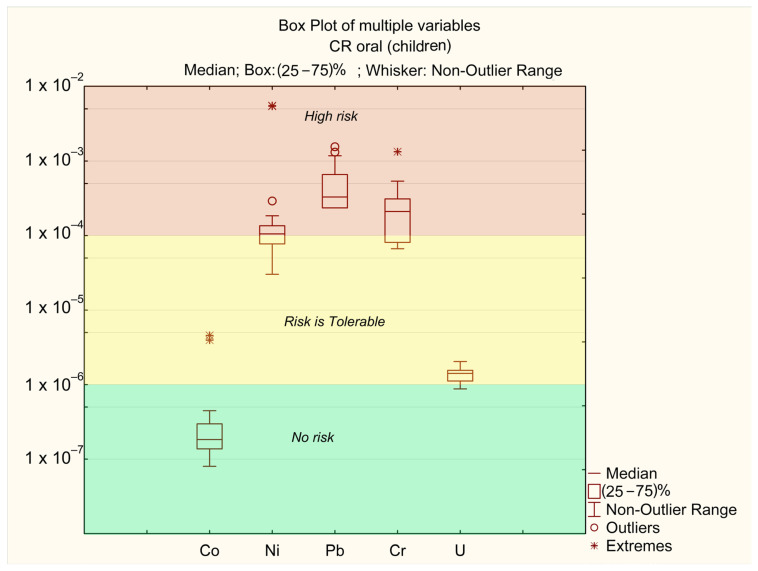
Oral CFi for children.

**Figure 9 ijerph-22-00560-f009:**
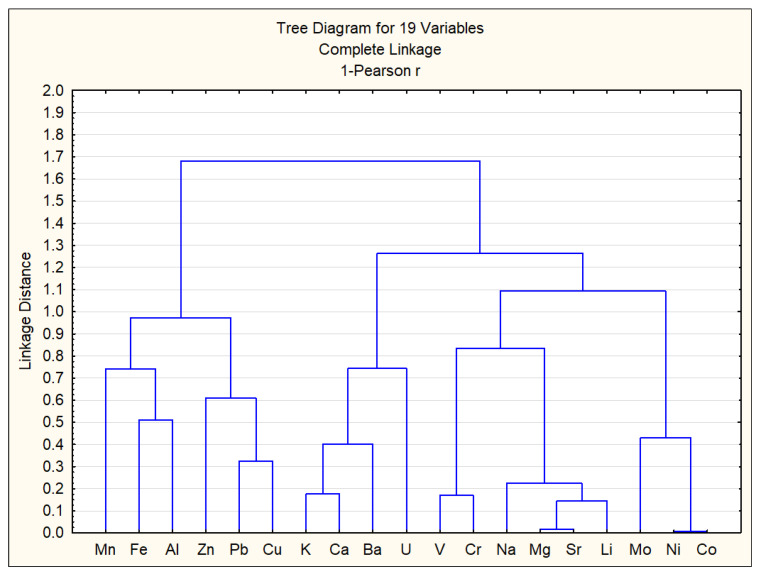
Cluster analysis of chemical elements.

**Figure 10 ijerph-22-00560-f010:**
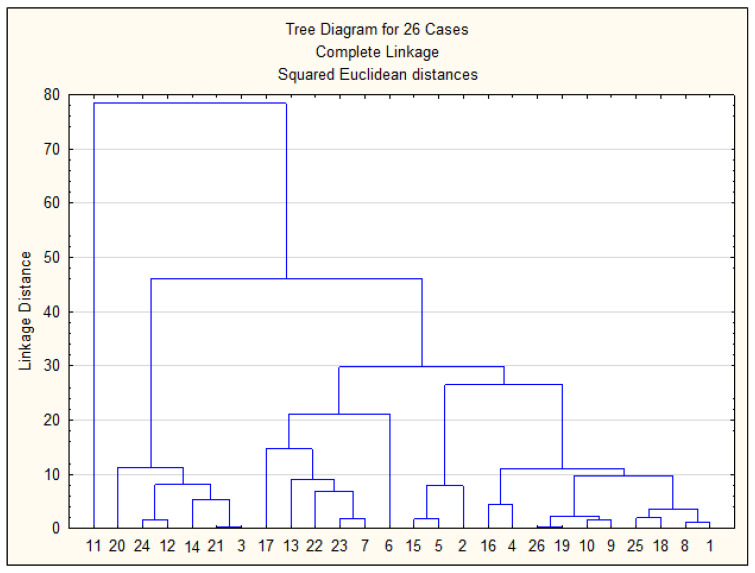
Cluster analysis of studied samples.

**Figure 11 ijerph-22-00560-f011:**
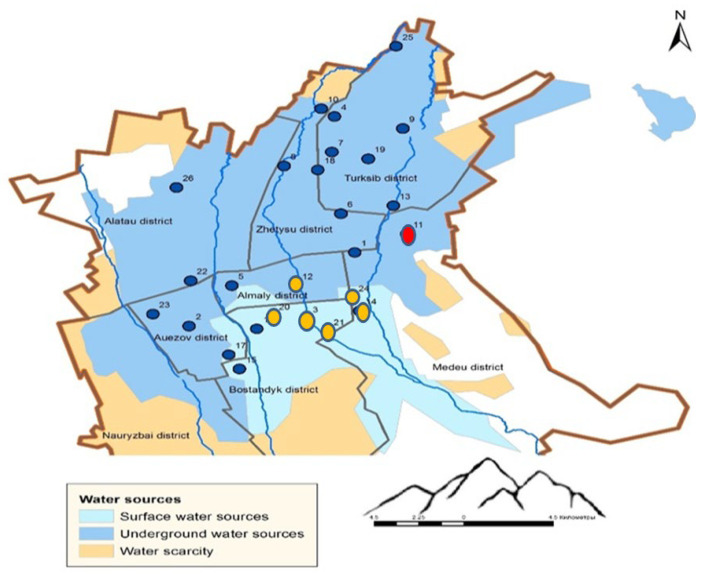
Sampling point classification into clusters.

**Table 2 ijerph-22-00560-t002:** Mean seasonal concentrations and standard deviations of chemical elements in drinking water samples of Almaty.

Sample Code.	Co.	Cu.	Li.	Mo.	Ni.	Pb.	U.	Al.	Ba.	Cr…
	Mean ± SD, μg/L									
1	0.36 ± 0.20	1.27 ± 1.03	3.41 ± 0.66	4.78 ± 1.28	4.15 ± 2.57	0.07 ± 0.04	11.1 ± 3.4	<3.00	33.8 ± 5.9	1.96 ± 1.26
2	0.24 ± 0.11	0.71 ± 0.46	3.19 ± 0.52	3.04 ± 0.17	3.14 ± 0.70	<0.05	12.6 ± 6.8	<3.00	59.1 ± 4.5	2.23 ± 0.47
3	<0.07	1.03 ± 0.61	0.79 ± 0.26	5.63 ± 2.65	1.59 ± 1.12	<0.05	8.7 ± 6.8	24.5 ± 14.0	30.8 ± 5.8	<0.70
4	<0.07	0.74 ± 0.31	3.10 ± 1.58	7.85 ± 0.91	0.68 ± 0.32	<0.05	9.6 ± 3.0	4.63 ± 1.48	27.5 ± 1.6	14.0 ± 16.6
5	0.13 ± 0.06	2.60 ± 0.81	1.88 ± 0.15	3.50 ± 0.67	2.39 ± 0.58	<0.05	13.8 ± 1.4	5.90 ± 3.87	46.1 ± 3.1	0.85 ± 0.26
6	3.46 ± 5.78	6.40 ± 5.19	3.00 ± 1.40	4.57 ± 1.17	122 ± 207	0.23 ± 0.31	16.7 ± 3.9	13.5 ± 7.3	34.4 ± 3.7	3.02 ± 2.01
7	0.17 ± 0.17	4.59 ± 3.81	2.34 ± 1.21	6.09 ± 1.05	2.05 ± 0.55	0.20 ± 0.13	10.7 ± 7.9	5.70 ± 4.68	29.8 ± 3.7	1.93 ± 1.68
8	0.26 ± 0.22	1.27 ± 1.14	3.31 ± 0.59	5.66 ± 0.57	1.75 ± 0.39	0.07 ± 0.03	12.7 ± 8.7	<3.00	29.6 ± 1.3	2.86 ± 0.74
9	0.20 ± 0.22	1.15 ± 0.66	2.80 ± 0.51	5.45 ± 0.37	1.73 ± 0.53	<0.05	16.5 ± 8.2	<3.00	27.7 ± 0.7	2.74 ± 0.62
10	0.14 ± 0.11	4.60 ± 3.98	3.00 ± 1.67	6.87 ± 1.38	1.64 ± 1.04	0.14 ± 0.08	15.9 ± 4.9	<3.00	26.2 ± 12.7	3.55 ± 0.74
11	4.00 ± 6.00	5.27 ± 2.96	3.95 ± 0.75	17.2 ± 23.2	124 ± 210	0.10 ± 0.08	14.6 ± 7.1	3.74 ± 1.28	35.3 ± 4.2	3.26 ± 2.17
12	0.12 ± 0.08	2.19 ± 1.27	0.68 ± 0.14	5.50 ± 2.87	6.55 ± 9.67	0.07 ± 0.03	13.1 ± 5.5	19.9 ± 13.8	31.2 ± 9.2	<0.70
13	0.39 ± 0.14	6.33 ± 5.04	3.02 ± 0.05	6.11 ± 0.13	3.32 ± 1.76	0.33 ± 0.16	8.8 ± 3.7	25.6 ± 11.0	31.3 ± 2.7	3.91 ± 0.54
14	0.09 ± 0.04	6.83 ± 4.08	0.79 ± 0.23	5.74 ± 2.39	1.74 ± 1.11	0.17 ± 0.16	8.6 ± 3.2	14.6 ± 1.83	20.0 ± 10.2	<0.70
15	0.14 ± 0.12	3.32 ± 1.97	2.26 ± 0.74	4.86 ± 0.51	2.89 ± 1.22	0.07 ± 0.03	13.9 ± 2.4	<3.00	47.6 ± 15.8	2.09 ± 0.26
16	0.14 ± 0.12	1.61 ± 1.21	1.67 ± 1.54	6.69 ± 1.23	1.93 ± 1.52	0.06 ± 0.02	11.1 ± 4.0	10.7 ± 6.9	34.3 ± 6.9	2.27 ± 2.73
17	0.15 ± 0.11	5.63 ± 4.38	2.33 ± 0.32	5.01 ± 0.98	2.17 ± 0.14	0.25 ± 0.11	18.6 ± 9.5	5.52 ± 2.90	41.5 ± 6.5	2.19 ± 0.13
18	0.34 ± 0.07	2.80 ± 0.35	2.76 ± 0.33	6.06 ± 0.27	2.35 ± 0.72	0.13 ± 0.11	9.0 ± 4.7	<3.00	33.6 ± 3.1	3.55 ± 0.51
19	0.21 ± 0.12	4.33 ± 1.00	2.47 ± 0.50	5.71 ± 0.26	2.29 ± 0.57	0.07 ± 0.03	13.7 ± 2.0	3.73 ± 1.27	28.8 ± 3.1	2.68 ± 0.32
20	<0.07	4.57 ± 2.64	0.60 ± 0.15	5.41 ± 2.51	1.67 ± 0.98	0.28 ± 0.40	14.2 ± 7.1	22.7 ± 6.8	29.0 ± 4.5	<0.70
21	<0.07	1.78 ± 0.70	0.56 ± 0.14	5.73 ± 2.72	1.38 ± 0.93	<0.05	8.0 ± 0.6	24.4 ± 13.1	28.5 ± 6.1	<0.70
22	0.16 ± 0.04	6.76 ± 6.39	1.55 ± 0.02	3.66 ± 1.54	2.51 ± 0.75	<0.05	14.5 ± 0.7	7.72 ± 2.64	47.0 ± 6.4	0.87 ± 0.23
23	0.19 ± 0.10	1.15 ± 0.81	2.23 ± 0.36	6.06 ± 0.18	2.51 ± 0.78	<0.05	12.0 ± 2.9	<3.00	40.8 ± 7.7	2.16 ± 0.38
24	0.34 ± 0.42	3.02 ± 2.71	1.20 ± 1.06	6.07 ± 1.22	3.05 ± 2.53	<0.05	13.7 ± 7.9	12.8 ± 11.0	30.0 ± 5.2	<0.70
25	0.12 ± 0.08	1.73 ± 0.43	3.37 ± 0.07	7.33 ± 0.36	2.78 ± 1.74	<0.05	10.2 ± 3.1	<3.00	36.5 ± 6.3	5.64 ± 0.29
26	0.16 ± 0.16	1.43 ± 0.97	2.43 ± 0.04	5.49 ± 0.61	2.60 ± 1.27	0.12 ± 0.12	14.0 ± 7.0	3.68 ± 1.17	29.4 ± 3.5	5.26 ± 3.06
Detection limit	0.07	0.5	0.2	0.3	0.5	0.05	0.03	3.0	0.5	0.7
Hazard class valid in Kazakhstan	2	3	2	2	3	2	1	2	2	3
MAC valid in Kazakhstan	100	1000	30	250	100	30	-	500	700	50
MAC USEPA	100		-		100	15	30	-	2000	
MAC WHO	-		-	70	70	10	30	-	700	
**Sample Code**	**Fe**	**Mn**	**Sr**	**V**	**Zn**	**Ca**	**K**	**Mg**	**Na**	**HMI**
	**Mean ± SD, μg/L**					**Mean ± SD, mg/L**				
**1**	4.91 ± 1.95	0.69 ± 0.33	528 ± 108	1.36 ± 0.62	40.9 ± 54.5	70.9 ± 9.3	1.97 ± 0.08	16.1 ± 1.7	20.4 ± 0.4	0.74
**2**	1.33 ± 0.80	<0.50	569 ± 14	<1.00	6.0 ± 3.6	70.9 ± 8.8	3.13 ± 0.13	13.5 ± 0.7	9.5 ± 0.9	0.76
**3**	20.2 ± 17.6	2.36 ± 2.65	109 ± 19	<1.00	2.0 ± 0.4	22.4 ± 3.6	1.01 ± 0.28	2.17 ± 0.52	3.7 ± 0.8	0.47
**4**	19.6 ± 31.5	1.42 ± 1.60	184 ± 5	3.68 ± 0.26	8.4 ± 12.7	20.4 ± 0.6	1.65 ± 0.10	3.72 ± 0.16	9.9 ± 0.7	0.58
**5**	3.26 ± 3.17	<0.50	293 ± 23	<1.00	6.6 ± 0.4	56.0 ± 5.9	2.15 ± 0.24	6.13 ± 0.94	10.8 ± 0.8	0.71
**6**	20.7 ± 17.4	15.7 ± 24.9	450 ± 121	1.24 ± 0.41	145 ± 223	51.4 ± 5.1	1.98 ± 0.16	12.2 ± 3.4	14.3 ± 5.0	0.93
**7**	10.2 ± 10.9	1.05 ± 0.53	366 ± 229	1.39 ± 0.68	44.1 ± 55.8	36.1 ± 22.0	1.69 ± 0.49	9.74 ± 6.50	13.0 ± 8.9	0.64
**8**	1.94 ± 1.29	<0.50	493 ± 48	1.17 ± 0.29	7.4 ± 6.0	44.4 ± 17.7	1.83 ± 0.12	13.5 ± 1.0	16.2 ± 3.8	0.76
**9**	2.13 ± 1.70	<0.50	458 ± 76	<1.00	3.1 ± 2.6	46.4 ± 20.4	1.91 ± 0.22	13.2 ± 1.6	14.4 ± 4.3	0.85
**10**	7.18 ± 4.68	<0.50	422 ± 233	2.15 ± 0.59	6.3 ± 4.7	37.8 ± 17.6	1.78 ± 0.11	11.1 ± 6.6	16.9 ± 8.2	0.85
**11**	10.9 ± 12.2	1.63 ± 1.95	511 ± 47	1.13 ± 0.23	17.7 ± 17.6	84.5 ± 8.0	2.06 ± 0.18	16.1 ± 1.6	20.7 ± 5.3	0.97
**12**	12.4 ± 10.3	3.53 ± 4.85	105 ± 13	<1.00	5.8 ± 5.6	22.3 ± 3.6	0.98 ± 0.28	2.16 ± 0.41	3.5 ± 0.8	0.60
**13**	5.98 ± 1.10	<0.50	526 ± 6	1.91 ± 0.09	300 ± 71	56.0 ± 1.5	1.89 ± 0.20	14.1 ± 0.4	17.7 ± 2.1	0.69
**14**	14.5 ± 11.7	3.45 ± 2.38	119 ± 23	<1.00	9.4 ± 3.3	24.6 ± 5.3	0.86 ± 0.13	2.88 ± 0.98	4.7 ± 1.8	0.44
**15**	2.73 ± 1.80	<0.50	447 ± 68	1.24 ± 0.42	6.7 ± 5.7	66.9 ± 6.1	2.63 ± 0.11	12.0 ± 1.4	8.8 ± 1.5	0.74
**16**	15.1 ± 10.5	<0.50	297 ± 291	1.58 ± 0.57	7.5 ± 4.7	35.7 ± 17.7	1.37 ± 0.33	7.15 ± 7.67	11.2 ± 12.5	0.63
**17**	3.04 ± 3.23	<0.50	415 ± 18	1.55 ± 0.56	3.6 ± 2.6	52.0 ± 18.6	2.55 ± 0.09	11.1 ± 0.3	8.6 ± 1.3	0.90
**18**	5.24 ± 1.74	<0.50	550 ± 8	1.41 ± 0.58	8.5 ± 2.3	56.7 ± 2.5	1.96 ± 0.15	14.4 ± 0.3	19.4 ± 0.6	0.65
**19**	2.89 ± 1.06	<0.50	478 ± 40	1.37 ± 0.64	14.1 ± 2.9	55.6 ± 3.0	1.91 ± 0.13	13.0 ± 1.4	14.8 ± 0.4	0.76
**20**	10.0 ± 8.50	2.21 ± 2.48	104 ± 21	<1.00	17.4 ± 24.1	21.3 ± 3.9	0.98 ± 0.30	2.06 ± 0.55	3.6 ± 0.7	0.64
**21**	8.15 ± 8.18	2.72 ± 3.84	102 ± 22	<1.00	5.8 ± 1.0	21.7 ± 5.0	0.95 ± 0.31	2.18 ± 0.64	3.5 ± 1.1	0.43
**22**	4.88 ± 3.48	0.68 ± 0.25	279 ± 10	<1.00	11.2 ± 7.0	54.1 ± 0.7	2.04 ± 0.04	5.54 ± 0.04	9.6 ± 2.1	0.72
**23**	2.65 ± 1.38	<0.50	402 ± 29	1.70 ± 0.61	8.7 ± 1.2	63.8 ± 2.6	2.46 ± 0.10	11.3 ± 0.4	8.7 ± 1.5	0.67
**24**	5.15 ± 4.38	24.1 ± 30.8	253 ± 251	<1.00	63.8 ± 96.3	37.6 ± 27.3	1.24 ± 0.51	6.46 ± 7.44	9.0 ± 9.8	0.68
**25**	5.13 ± 2.43	1.74 ± 1.75	451 ± 1	3.88 ± 0.43	7.1 ± 1.0	36.7 ± 4.1	1.55 ± 0.18	10.9 ± 1.01	27.2 ± 4.0	0.74
**26**	4.91 ± 1.95	<0.50	502 ± 79	1.98 ± 0.49	80.7 ± 83.7	49.5 ± 14.8	1.96 ± 0.10	13.6 ± 1.2	15.6 ± 5.7	0.77
**Detection limit**	0.4	0.5	0.5	1.0	0.7	0.01	0.015	0.03	0.01	
**Hazard class valid in Kazakhstan**	**3**	**3**	**2**	**3**	**3**			**3**	**2**	
**MAC valid in Kazakhstan**	**300**	**100**	**7000**	**100**	**5000**	**30**		**20**	**200**	
**MAC USEPA**	**300**	**50**	**-**	**-**	**5000**	**-**		**-**	**-**	
**MAC WHO**	**-**	**400**	**-**	**-**	**5000**	**-**		**-**	**200**	

**Table 3 ijerph-22-00560-t003:** Carcinogenic risk index (CR) calculation results for adults and children.

Sampling Point	Adults			Children		
	Dermal	Oral	CR	Dermal	Oral	CR
1	4.83 × 10^−8^	5.77 × 10^−4^	5.78 × 10^−4^	4.96 × 10^−8^	7.02 × 10^−4^	7.02 × 10^−4^
2	5.24 × 10^−8^	4.92 × 10^−4^	4.92 × 10^−4^	5.38 × 10^−8^	5.89 × 10^−4^	5.89 × 10^−4^
3	1.93 × 10^−8^	3.14 × 10^−4^	3.14 × 10^−4^	1.98 × 10^−8^	3.74 × 10^−4^	3.74 × 10^−4^
4	3.16 × 10^−7^	1.28 × 10^−3^	1.28 × 10^−3^	3.24 × 10^−7^	1.60 × 10^−4^	1.60 × 10^−3^
5	2.81 × 10^−8^	3.65 × 10^−4^	3.65 × 10^−4^	2.89 × 10^−8^	4.24 × 10^−4^	4.24 × 10^−4^
6	9.00 × 10^−8^	5.38 × 10^−3^	5.38 × 10^−3^	9.24 × 10^−8^	6.80 × 10^−4^	6.80 × 10^−3^
7	5.93 × 10^−8^	9.81 × 10^−4^	9.82 × 10^−4^	6.09 × 10^−8^	1.22 × 10^−4^	1.22 × 10^−3^
8	6.84 × 10^−8^	5.65 × 10^−4^	5.65 × 10^−4^	7.03 × 10^−8^	6.81 × 10^−4^	6.81 × 10^−4^
9	6.53 × 10^−8^	4.90 × 10^−4^	4.90 × 10^−4^	6.71 × 10^−8^	5.75 × 10^−4^	5.75 × 10^−4^
10	9.56 × 10^−8^	8.79 × 10^−4^	8.79 × 10^−4^	9.81 × 10^−8^	1.07 × 10^−3^	1.07 × 10^−3^
11	9.14 × 10^−8^	4.98 × 10^−3^	4.98 × 10^−3^	9.39 × 10^−8^	6.30 × 10^−3^	6.30 × 10^−3^
12	2.33 × 10^−8^	5.72 × 10^−4^	5.72 × 10^−4^	2.40 × 10^−8^	6.89 × 10^−4^	6.89 × 10^−4^
13	1.10 × 10^−7^	1.65 × 10^−3^	1.65 × 10^−3^	1.13 × 10^−7^	2.07 × 10^−3^	2.07 × 10^−3^
14	3.96 × 10^−8^	7.63 × 10^−4^	7.63 × 10^−4^	4.07 × 10^−8^	9.45 × 10^−4^	9.45 × 10^−4^
15	5.84 × 10^−8^	5.50 × 10^−4^	5.50 × 10^−4^	6.00 × 10^−8^	6.59 × 10^−4^	6.59 × 10^−4^
16	5.64 × 10^−8^	4.86 × 10^−4^	4.86 × 10^−4^	5.79 × 10^−8^	5.86 × 10^−4^	5.86 × 10^−4^
17	6.88 × 10^−8^	1.21 × 10^−3^	1.21 × 10^−3^	7.06 × 10^−8^	1.48 × 10^−3^	1.48 × 10^−3^
18	8.92 × 10^−8^	8.50 × 10^−4^	8.50 × 10^−4^	9.16 × 10^−8^	1.06 × 10^−3^	1.06 × 10^−3^
19	7.52 × 10^−8^	5.73 × 10^−4^	5.73 × 10^−4^	7.72 × 10^−8^	6.88 × 10^−4^	6.88 × 10^−4^
20	3.17 × 10^−8^	1.18 × 10^−3^	1.18 × 10^−3^	3.25 × 10^−8^	1.46 × 10^−3^	1.46 × 10^−3^
21	2.19 × 10^−8^	3.05 × 10^−4^	3.05 × 10^−4^	2.25 × 10^−8^	3.64 × 10^−4^	3.64 × 10^−4^
22	4.32 × 10^−8^	3.73 × 10^−4^	3.73 × 10^−4^	4.43 × 10^−8^	4.31 × 10^−4^	4.32 × 10^−4^
23	5.24 × 10^−8^	4.63 × 10^−4^	4.63 × 10^−4^	5.38 × 10^−8^	5.54 × 10^−4^	5.54 × 10^−4^
24	2.63 × 10^−8^	3.77 × 10^−4^	3.77 × 10^−4^	2.70 × 10^−8^	4.39 × 10^−4^	4.39 × 10^−4^
25	1.32 × 10^−7^	7.29 × 10^−4^	7.29 × 10^−4^	1.36 × 10^−7^	8.97 × 10^−4^	8.97 × 10^−4^
26	1.23 × 10^−7^	9.62 × 10^−4^	9.62 × 10^−4^	1.26 × 10^−7^	1.18 × 10^−3^	1.18 × 10^−3^
Max	3.16 × 10^−7^	5.38 × 10^−3^	5.38 × 10^−3^	3.24 × 10^−7^	6.80 × 10^−3^	6.80 × 10^−3^
Min	1.93 × 10^−8^	3.05 × 10^−4^	3.05 × 10^−4^	1.98 × 10^−8^	3.64 × 10^−4^	3.64 × 10^−4^

**Table 4 ijerph-22-00560-t004:** Pearson correlation matrix of elements.

	Co	Cu	Li	Mo	Ni	Pb	U	Al	Ba	Cr	Fe	Mn	Sr	V	Zn	Ca	K	Mg	Na
**Co**	1.00																		
**Cu**	0.37	1.00																	
**Li**	0.41	−0.07	1.00																
**Mo**	0.61	0.12	0.35	1.00															
**Ni**	0.99	0.37	0.36	0.57	1.00														
**Pb**	0.20	0.68	0.02	0.00	0.20	1.00													
**U**	0.31	0.27	0.20	−0.03	0.32	0.14	1.00												
**Al**	−0.04	0.22	−0.68	−0.10	−0.01	0.36	−0.35	1.00											
**Ba**	0.03	−0.10	0.20	−0.26	0.03	−0.24	0.26	−0.32	1.00										
**Cr**	0.04	−0.24	0.54	0.27	0.04	−0.06	−0.13	−0.34	−0.15	1.00									
**Fe**	0.33	0.13	−0.26	0.23	0.39	0.15	−0.28	0.49	−0.41	0.22	1.00								
**Mn**	0.32	0.17	−0.21	−0.03	0.34	0.03	0.16	0.26	−0.16	−0.16	0.26	1.00							
**Sr**	0.29	−0.01	0.86	0.11	0.22	0.09	0.28	−0.66	0.34	0.19	−0.54	−0.20	1.00						
**V**	−0.09	−0.19	0.49	0.23	−0.09	−0.01	−0.19	−0.32	−0.13	0.83	0.11	−0.18	0.21	1.00					
**Zn**	0.25	0.39	0.21	−0.03	0.23	0.62	−0.10	0.39	−0.13	0.10	0.09	0.25	0.28	0.10	1.00				
**Ca**	0.44	0.10	0.65	0.19	0.37	−0.04	0.33	−0.56	0.60	−0.06	−0.53	−0.16	0.81	−0.12	0.15	1.00			
**K**	0.16	−0.03	0.64	−0.11	0.12	−0.05	0.43	−0.67	0.76	0.15	−0.56	−0.27	0.76	0.07	0.04	0.82	1.00		
**Mg**	0.36	0.00	0.85	0.21	0.29	0.12	0.29	−0.63	0.24	0.17	−0.51	−0.18	0.98	0.17	0.28	0.81	0.70	1.00	
**Na**	0.32	−0.01	0.84	0.36	0.26	0.03	0.05	−0.56	−0.01	0.37	−0.28	−0.14	0.78	0.52	0.25	0.50	0.33	0.78	1.00

## Data Availability

Data Availability Statement: Our dataset has been assigned two important links: A reviewer URL: https://datadryad.org/stash/dataset/doi:10.5061/dryad.dfn2z359h (accessed on 14 February 2025); a unique digital object identifier (DOI): https://doi.org/10.5061/dryad.dfn2z359h (accessed on 14 February 2025).

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
