# Peer review of "Study of the Trace Element Composition of Drinking Water in Almaty City and Human Health Risk Assessment"

_ijerph, 2025, doi:10.3390/ijerph22040560_

Round 1

Reviewer 1 Report

Comments and Suggestions for Authors

The manuscript submitted by the authors, titled Study of the Trace Element Composition of Drinking Water in Almaty City and Human Health Risk Assessment, presents an interesting and relevant study. The research is well within the scope of the journal; however, the manuscript has several shortcomings that need to be addressed. A major revision is required. Below are the comments for the authors' attention. These questions aim to encourage the authors to provide deeper mechanistic explanations, compare their findings with existing studies, and address practical implementation challenges.

  1. “Seasonal average concentrations and standard deviations were computed for chemical elements including arsenic (As), beryllium (Be), cobalt (Co), cadmium (Cd), copper (Cu), lithium (Li), molybdenum (Mo), nickel (Ni), lead (Pb), selenium (Se), uranium (U), mercury (Hg), aluminum (Al), barium (Ba), chromium (Cr), iron (Fe), manganese (Mn), strontium (Sr), vanadium (V), zinc (Zn), calcium (Ca), potassium (K), magnesium (Mg), and sodium (Na) across three distinct data sets.”
    should be revised for better readability and grammatical accuracy.
  2. The manuscript should provide more details about the study area. Were the selected sampling points representative of the overall drinking water supply in Almaty? What criteria were used to choose them?
  3. Please clarify how significant the seasonal variations in heavy metal concentrations were. What factors contributed to these differences? Were any statistical tests performed to support these findings?
  4. The study should explicitly identify potential sources of heavy metal contamination. Were both natural and anthropogenic sources analyzed in detail?
  5. The authors should elaborate on how the thresholds for carcinogenic and non-carcinogenic risks were determined. Are there alternative risk assessment models that could yield different conclusions? This discussion should be expanded in the manuscript.
  6. Based on the findings, what specific water treatment strategies can be recommended to mitigate heavy metal exposure risks?
  7. If possible, the study should compare its findings with previous studiesconducted in Almaty or similar urban environments. This would help in identifying trends in water contamination over time.
  8. Given the identified health risks, what policy recommendations can be made to local authorities in Almaty? How can the study contribute to improving drinking water safety regulations?
  9. The authors should clearly outline the main sources of uncertainty in the study. How might these uncertainties have affected the risk assessment outcomes? A dedicated section discussing the limitations of the study is recommended.

Author Response

Dear Reviewer,

Thank you for your valuable feedback. We have carefully considered all your comments and suggestions, and we believe that incorporating them has significantly enhanced the quality and clarity of our article. These improvements make our study more accessible and beneficial to the readers.

We sincerely appreciate your input in helping us refine our work. Below, you will find detailed responses along with the corresponding revisions and corrections, all highlighted and tracked in the resubmitted files.

Reviewer 2 Report

Comments and Suggestions for Authors

SUMMARY

The study investigates the trace element composition of drinking water in Almaty, Kazakhstan, and assesses the associated human health risks. ​ Researchers collected 78 water samples from 26 locations across different seasons in 2023. ​ The analysis focused on elements like arsenic, beryllium, cobalt, cadmium, copper, lithium, molybdenum, nickel, lead, selenium, uranium, mercury, aluminum, barium, chromium, iron, manganese, strontium, vanadium, zinc, calcium, potassium, magnesium, and sodium. ​ The water quality was evaluated against national, international, and WHO standards. ​

The study found that, except for two sampling points, non-carcinogenic risks were below acceptable thresholds, with oral ingestion being the primary exposure route. ​ Carcinogenic risks from nickel, lead, and chromium were identified, with most sites falling into the "high risk" category. ​ The highest contributors to heavy metal contamination were uranium, sodium, and lithium. ​

The research employed Inductively Coupled Plasma Mass Spectrometry (ICP-MS) and Optical Emission Spectrometry (ICP-OES) for elemental analysis. ​ Quality control measures included duplicate samples, blank samples, and certified reference materials. ​ The study also used Pearson correlation and cluster analysis to identify relationships between elements and potential pollution sources. ​

Results indicated that while most elements did not exceed maximum acceptable concentrations (MAC), nickel and cobalt showed high seasonal variability, particularly in summer. ​ The study highlighted the need for improved water purification and ongoing monitoring to mitigate health risks. ​ The primary sources of contamination were attributed to the corrosion of water supply pipes. ​

The study concludes that while most water samples met safety standards, certain areas showed elevated risks, necessitating attention from authorities to address potential health impacts from heavy metals in drinking water. The findings underscore the importance of continuous surveillance and advanced purification methods to ensure public health safety. ​

STRENGTHS

The research on the trace element composition of drinking water in Almaty, Kazakhstan, has several strengths:

  1. Comprehensive Sampling: The study collected 78 water samples from 26 locations across different seasons, providing a robust dataset that captures seasonal variations in water quality. ​
  2. Advanced Analytical Techniques: The use of Inductively Coupled Plasma Mass Spectrometry (ICP-MS) and Optical Emission Spectrometry (ICP-OES) ensures precise and accurate measurement of trace elements. ​
  3. Quality Control: The research includes rigorous quality control measures, such as duplicate samples, blank samples, and certified reference materials, ensuring the reliability of the results. ​
  4. Health Risk Assessment: The study not only measures the concentrations of various elements but also assesses the associated non-carcinogenic and carcinogenic health risks, providing a comprehensive understanding of potential health impacts. ​
  5. Comparison with Standards: The water quality is evaluated against national, international, and WHO standards, ensuring that the findings are relevant and actionable for policymakers. ​
  6. Identification of Contamination Sources: The research identifies potential sources of contamination, such as the corrosion of water supply pipes, which is crucial for developing targeted mitigation strategies. ​
  7. Statistical Analysis: The use of Pearson correlation and cluster analysis helps in understanding the relationships between different elements and their sources, adding depth to the findings. ​
  8. Public Health Implications: The study highlights areas with elevated health risks, emphasizing the need for improved water purification and ongoing monitoring, which is vital for protecting public health. ​
  9. Seasonal Variation Analysis: By analyzing samples across different seasons, the study provides insights into how water quality changes over time, which is important for developing effective water management strategies.

Overall, the research is thorough, methodologically sound, and provides valuable insights into the water quality and associated health risks in Almaty. ​

KEY FINDINGS

The main findings regarding heavy metal concentrations in the drinking water of Almaty are as follows:

  1. General Compliance: For most sampling points, the concentrations of heavy metals categorized under hazard classes 3 and 4 did not exceed the Maximum Acceptable Concentrations (MAC) set by national and international standards. ​
  2. Nickel and Cobalt Variability: High seasonal variability was observed for nickel (Ni) and cobalt (Co), particularly during the summer. ​ At sampling points 6 and 11, nickel concentrations exceeded the MAC by more than three times, reaching levels of 361 µg/L and 366 µg/L, respectively. ​
  3. Uranium, Sodium, and Lithium: These elements were significant contributors to the Heavy Metal Index (HMI). ​ Uranium concentrations were consistently elevated across all sampling points, ranging from 8.0 µg/L to 18.6 µg/L. ​
  4. Localized Contamination: Certain areas showed localized seasonal contamination, particularly in the summer, indicating potential sources of pollution affecting specific parts of the water supply. ​
  5. Heavy Metal Index (HMI): While most sampling points had HMI values below 1, indicating compliance with safety standards, points 6, 11, and 17 had HMI values approaching 1, suggesting relatively lower water quality in terms of heavy metal contamination. ​
  6. Carcinogenic Risks: The study identified high carcinogenic risks associated with nickel (Ni), lead (Pb), and chromium (Cr) at most sampling sites, with risk values categorically classified within the "high risk" designation. ​
  7. Non-Carcinogenic Risks: Except for two sampling points, the levels of non-carcinogenic risk remained below the acceptable threshold. ​ The primary exposure route for both adults and children was oral ingestion of hazardous elements. ​

These findings highlight the need for improved water purification and ongoing monitoring to mitigate health risks associated with heavy metal contamination in Almaty's drinking water. ​

PUBLIC HEALTH

The public health findings from the study on the trace element composition of drinking water in Almaty are significant and highlight several areas of concern:

  1. Carcinogenic Risks: The study identified high carcinogenic risks associated with the presence of nickel (Ni), lead (Pb), and chromium (Cr) in the drinking water. ​ The carcinogenic risk values for these elements at most sampling sites were classified within the "high risk" designation, indicating a serious potential for long-term health impacts such as cancer. ​
  2. Non-Carcinogenic Risks: Except for two sampling points, the levels of non-carcinogenic risk remained below the acceptable threshold (HI < 1). ​ However, at points 6 and 11, the hazard index (HI) values exceeded 1, indicating a low risk for non-carcinogenic effects. ​ The primary exposure route for both adults and children was oral ingestion of toxic elements. ​
  3. Elevated Uranium Levels: Uranium concentrations were consistently elevated across all sampling points, contributing significantly to the Heavy Metal Index (HMI). ​ Although the concentrations did not exceed the MAC, the presence of uranium in drinking water is concerning due to its potential health impacts.
  4. Localized Contamination: Certain areas showed localized seasonal contamination, particularly in the summer, with high concentrations of nickel and cobalt. ​ This suggests that specific parts of the water supply are more vulnerable to contamination, potentially due to factors like pipeline corrosion or localized pollution sources. ​
  5. Need for Improved Water Purification: The findings underscore the necessity for enhanced water purification methodologies to reduce the levels of hazardous elements in the drinking water. ​ This is crucial for protecting public health, especially in areas with identified high risks. ​
  6. Ongoing Surveillance: The study highlights the importance of ongoing monitoring and surveillance of drinking water quality to detect and address contamination issues promptly. ​ Regular monitoring can help in mitigating health risks and ensuring the safety of the water supply. ​
  7. Health Impacts of Heavy Metals: Long-term exposure to elevated concentrations of heavy metals like nickel, lead, and chromium can lead to serious health issues, including immunological effects, developmental delays, neurotoxicity, and gastrointestinal problems. The study emphasizes the need for public health interventions to minimize these risks. ​

Overall, the public health findings from this study call for immediate action to improve water quality and protect the health of the residents of Almaty. ​ Enhanced purification processes, regular monitoring, and targeted interventions in high-risk areas are essential steps to mitigate the identified health risks. ​

WEAKNESESS

The study on the trace element composition of drinking water in Almaty has several major weaknesses that should be considered:

  1. Seasonal Variability: The study only covers three seasons (winter, summer, and autumn) of 2023. This limited timeframe may not capture the full range of seasonal variations and potential long-term trends in heavy metal concentrations.
  2. Localized Sampling: The study focuses on 26 sampling points within Almaty. ​ While these points represent various districts, the limited number of sampling locations may not provide a comprehensive overview of the entire city's water quality. ​
  3. Methodological Uncertainties: The study acknowledges uncertainties in the applied methodology, including generalized calculations of chronic daily intake (CDI) and the lack of precise values for reference doses (RfD) and slope factors (SF). ​ These uncertainties can affect the accuracy of the health risk assessments. ​
  4. Potential Sources of Contamination: The study suggests that the most likely source of heavy metal contamination is the corrosion of water supply pipes. ​ However, it does not provide detailed investigations or evidence to confirm this hypothesis, leaving the exact sources of contamination unclear.
  5. Limited Scope of Elements: While the study analyzes a wide range of heavy metals, it does not include all possible contaminants that could affect water quality. Other potentially harmful substances may be present but were not assessed. ​
  6. Lack of Longitudinal Data: The study is based on data collected over a single year. Longitudinal studies over multiple years would provide a more robust understanding of trends and variations in water quality.
  7. Absence of Human Health Data: The study relies on calculated risk assessments rather than direct health data from the population. ​ Actual health outcomes related to heavy metal exposure in Almaty residents are not examined, which limits the ability to correlate water quality with health impacts directly.
  8. Generalized Risk Assessment: The risk assessment is based on standard models and assumptions, which may not fully account for local factors such as dietary habits, water consumption patterns, and individual susceptibilities.
  9. Funding and Potential Bias: The study was funded by the Ministry of Science and Higher Education of the Republic of Kazakhstan. ​ While there is no indication of bias, funding sources can sometimes influence study outcomes, and this potential conflict of interest should be acknowledged.

Addressing these weaknesses in future research could provide a more comprehensive and accurate assessment of drinking water quality and its health impacts in Almaty.

OVERALL

The research on the trace element composition of drinking water in Almaty is a strong and valuable contribution, providing essential information to both the public and the scientific community. ​ By analyzing 78 drinking water samples from 26 locations across Almaty during the winter, summer, and autumn of 2023, the study offers a comprehensive assessment of water quality. ​ It identifies the concentrations of various heavy metals and other elements, highlighting specific areas with elevated levels of contaminants like nickel (Ni), lead (Pb), and chromium (Cr). ​

The health risk assessment included in the study is particularly significant, as it evaluates both carcinogenic and non-carcinogenic risks associated with these elements. ​ This assessment is crucial for understanding the potential long-term health impacts on the population. ​ The findings underscore the need for enhanced water purification methodologies and ongoing surveillance to protect public health. ​The research employs advanced analytical techniques and robust statistical methods, ensuring the accuracy and reliability of the data. ​ It also provides actionable insights and recommendations for policymakers and public health officials to mitigate the risks associated with heavy metal contamination. ​

Overall, this research is a significant and valuable contribution that enhances the understanding of water quality issues in urban environments and guides efforts to ensure safe drinking water for the population. ​

Author Response

(The authors gave the same response as above.)

Reviewer 3 Report

Comments and Suggestions for Authors

Abstract

  • change forchemical to for chemical
  • Put space between WHO standards. Drinking
  • Recommend changing Drinking water pollution to Drinking water contaminants
  • No substantial differences in

Introduction

  • Do you wish to state that EPA’s lifetime health advisories are based on a 70 kilogram person drinking 2 liters of water per day over a lifetime?

Study Area

  • For the study area, there is a discussion on the sources of water in Almaty (mountain rivers, lakes, and ground water). Would you also state briefly the types of water plants in the area and what treatment technologies are utilized (e.g., conventional filtration, ground water treated with disinfection only, membranes, or advanced selected media filters)

Material and Methods

  • State that samples collected were finished drinking water samples that were collected within the distribution system at residences and large building complexes.
  • What was the flushing time to flush the tap prior to obtaining a sample? Was a calculated flush time used based on the pipe diameters from the main to the tap in the residence/building?
  • Were the samples preserved at the time of collection with nitric acid? There is the mention that the sample containers were washed with sample water and nitric acid.
  • What was the calibration frequency for the ICP spectrometers?

Results and Discussions

  • In the Materials and Methods section, it states the following “private residences and multi-apartment complexes”. Recommend being consistent in this section by using the same instead of private and multi-apartment edifices.
  • During the summer of 2023 that resulted in high Ni and Co levels, was the area in a drought condition or elevated temperatures occurred?
  • For Figure 2, with most of the higher HMI occurring in the Northern part of the sampling area, could this is attributed to older infrastructure? Could you please discuss the material composition of these distribution system water lines?
  • Any recommendations that the end user can take to protect their drinking water like a point of use device, filter, or letting the water settle out prior to drinking?

Overall Things to Consider

  • In Table 2 there are some traces of lead detected. Could this be explained as an issue with the premise plumbing fixture or an indication of lead plumbing components somewhere in the distribution system?
  • For some of the trace element detections, could this be attributed to any mining or industrial activities near the source waters. Example lithium?
  • From the trace element data, could any co occurrences be identified if certain elements are associated with others?

Author Response

(The authors gave the same response as above.)

Reviewer 4 Report

Comments and Suggestions for Authors

This manuscript is well organized and well written. The authors provide and summarize helpful toxicological and risk assessment data on elemental composition, including heavy metals and essential metals levels detected at the Almaty City drinking water system, and comparisons are made against local and international health advisory standards. The following are several revision suggestions for the authors’ considerations:

Introduction section, page 2: Explain what is meant by “evolutionary” factors. Also provide reference source citation.

Study area: Where applicable/ available, provide reference source citation on the climate and geographical information of the study area.

Section 3.1, Sampling:

Clarify if the nitric acid was added as sample preservative or used in cleaning procedure, since rinsing the bottle with the sample, before collecting it would wash out the nitric acid, if used for the purpose of preserving the collected water samples.

Specify analytical methods utilized or refer the reader to the proceeding section 3.2.

Provide specific details on the systematic approach of sampling.

Section 3.6: Statistical analyses should be described not just in general terms, rather, explain clearly how specific analysis methods and relative comparative quantitative and qualitative analysis were applied to this data.

Table 2: Specify sample sizes for the presented means of the analytical data. Explain how water samples with results below detection limits were handled in calculating the means. On the column heading for the listed elements, specify the analytical detection limit for each element.

Figures 3 and 4: The box plots are difficult to see where the scale of measurements are low. Having a dual scale axis or zoomed-in plots would make it clearer to compare the measured levels of the different elements.

Table 6: Explain what the highlighted segments represent.

Figures 7 and 8: Explain what is meant by “ non-outlier range”. How were outliers identified and treated?

Page 19, last paragraph, before section 4.3: Specify which toxic element(s) are being referred to.

Section 4.3, first paragraph is more fitting content for methods and materials section.

Conclusion section: This section can be shortened to exclude repeated detailed information from the results section. Instead, add contents about cautionary points and limitations on interpreting the results and how the limitations can be addressed in future studies. Also explain how far apart were the seasonal data collections and how this could impact the waste quality outcomes. Additionally, when discussing significance of risks for cobalt, it should be clarified what statistical analysis, including level of significance, were utilized to compare the carcinogenic risks between children and adults.

Author Response

(The authors gave the same response as above.)

Round 2

Reviewer 1 Report

Comments and Suggestions for Authors

The authors have addressed all the comments satisfactorily.

Author Response

Thank you very much for your positive evaluation. I sincerely appreciate your valuable comments and suggestions — all of them have been carefully considered and addressed by the authors in the revised version of the manuscript.

Reviewer 4 Report

Comments and Suggestions for Authors

The revised manuscript is notably improved. The authors have reasonably addressed the noted review points from the original manuscript.

It is noted that Response 9 presented in the authors’ cover letter does not appear to have been changed in the revised manuscript.

Author Response

Thank you very much for your constructive feedback and for acknowledging the improvements in the revised manuscript. We apologize for the oversight regarding Response 9. We have now made the corresponding changes in the manuscript to ensure it aligns with our response in the cover letter.